# Reducing the Rank of Relational Factorization Models by Including Observable Patterns

**Maximilian Nickel**[1,2]        **Xueyan Jiang**[3,4]        **Volker Tresp**[3,4]

[1]LCSL, Poggio Lab, Massachusetts Institute of Technology, Cambridge, MA, USA
[2]Istituto Italiano di Tecnologia, Genova, Italy
[3]Ludwig Maximilian University, Munich, Germany
[4]Siemens AG, Corporate Technology, Munich, Germany
`mnick@mit.edu, {xueyan.jiang.ext,volker.tresp}@siemens.com`

## Abstract

Tensor factorization has become a popular method for learning from multi-relational data. In this context, the rank of the factorization is an important parameter that determines runtime as well as generalization ability. To identify conditions under which factorization is an efficient approach for learning from relational data, we derive upper and lower bounds on the rank required to recover adjacency tensors. Based on our findings, we propose a novel additive tensor factorization model to learn from latent and observable patterns on multi-relational data and present a scalable algorithm for computing the factorization. We show experimentally both that the proposed additive model does improve the predictive performance over pure latent variable methods and that it also reduces the required rank — and therefore runtime and memory complexity — significantly.

## 1 Introduction

Relational and graph-structured data has become ubiquitous in many fields of application such as social network analysis, bioinformatics, and artificial intelligence. Moreover, relational data is generated in unprecedented amounts in projects like the Semantic Web, YAGO [27], NELL [4], and Google's Knowledge Graph [5] such that learning from relational data, and in particular learning from large-scale relational data, has become an important subfield of machine learning. Existing approaches to relational learning can approximately be divided into two groups: First, methods that explain relationships via observable variables, i.e. via the observed relationships and attributes of entities, and second, methods that explain relationships via a set of latent variables. The objective of latent variable models is to infer the states of these hidden variables which, once known, permit the prediction of unknown relationships. Methods for learning from observable variables cover a wide range of approaches, e.g. inductive logic programming methods such as FOIL [23], statistical relational learning methods such as Probabilistic Relational Models [6] and Markov Logic Networks [24], and link prediction heuristics based on the Jaccard's Coefficient and the Katz Centrality [16]. Important examples of latent variable models for relational data include the IHRM and the IRM [29, 10], the Mixed Membership Stochastic Blockmodel [1] and low-rank matrix factorizations [16, 26, 7]. More recently, tensor factorization, a generalization of matrix factorization to higher-order data, has shown state-of-the-art results for relationship prediction on *multi-relational* data [21, 8, 2, 13]. The number of latent variables in tensor factorization is determined via the number of latent components used in the factorization, which in turn is bounded by the factorization rank. While tensor and matrix factorization algorithms scale typically well with the size of the data — which is one reason for their appeal — they often do not scale well with respect to the rank of the factorization. For instance, RESCAL is a state-of-the art relational learning method based on tensor factorization which can be applied to large knowledge bases consisting of millions of entities and billions of known facts [22].

However, while the runtime of the most scalable known algorithm to compute RESCAL scales linearly with the number of entities, linearly with the number of relations, and linearly with the number of known facts, it scales *cubical* with regard to the rank of the factorization [22].[1] Moreover, the memory requirements of tensor factorizations like RESCAL become quickly infeasible on large data sets if the factorization rank is large and no additional sparsity of the factors is enforced. Hence, tensor (and matrix) rank is a central parameter of factorization methods that determines generalization ability as well as scalability. In this paper we study therefore how the rank of factorization methods can be reduced while maintaining their predictive performance and scalability. We first analyze under which conditions tensor and matrix factorization requires high or low rank on relational data. Based on our findings, we then propose an additive tensor decomposition approach to reduce the required rank of the factorization by combining latent and observable variable approaches.

This paper is organized as follows: In section 2 we develop the main theoretical results of this paper, where we show that the rank of an adjacency tensor is lower bounded by the maximum number of strongly connected components of a single relation and upper bounded by the sum of diclique partition numbers of all relations. Based on our theoretical results, we propose in section 3 a novel tensor decomposition approach for multi-relational data and present a scalable algorithm to compute the decomposition. In section 4 we evaluate our model on various multi-relational datasets.

**Preliminaries** We will model relational data as a directed graph (digraph), i.e. as an ordered pair $\Gamma = (\mathcal{V}, \mathcal{E})$ of a nonempty set of vertices $\mathcal{V}$ and a set of directed edges $\mathcal{E} \subseteq \mathcal{V} \times \mathcal{V}$. An existing edge between node $v_i$ and $v_j$ will be denoted by $v_i \rightsquigarrow v_j$. By a slight abuse of notation, $\Gamma(Y)$ will indicate the digraph $\Gamma$ associated with an adjacency matrix $Y \in \{0,1\}^{N \times N}$. Next, we will briefly review further concepts of tensor and graph theory that are important for the course of this paper.

**Definition 1.** *A strongly connected component of a digraph $\Gamma$ is a maximal subgraph $\Psi$ for which every vertex is reachable from any other vertex in $\Psi$ by following the directional edges in the subgraph. A strongly connected component is* trivial *if it consists only of a single element, i.e. if it is of the form $\Psi = (\{v_i\}, \varnothing)$, and* nontrivial *otherwise.*

We will denote the number of strongly connected components in a digraph $\Gamma$ by $\mathrm{scc}(\Gamma)$. The number of nontrivially connected components will be denoted by $\mathrm{scc}_+(\Gamma)$.

**Definition 2.** *A digraph $\Gamma = (\mathcal{V}, \mathcal{E})$ is a diclique if it is an orientation of a complete undirected bipartite graph with bipartition $(\mathcal{V}_1, \mathcal{V}_2)$ such that $v_1 \in \mathcal{V}_1$ and $v_2 \in \mathcal{V}_2$ for every edge $v_1 \rightsquigarrow v_2 \in \mathcal{E}$.*

Figure 3 in supplementary material A shows an example of a diclique. Please note that dicliques consist only of trivially strongly connected components, as there cannot exist any cycles in a diclique. Given the concept of a diclique, the diclique partitioning number of a digraph is defined as:

**Definition 3.** *The diclique partition number $\mathrm{dp}(\Gamma)$ of a digraph $\Gamma = (\mathcal{V}, \mathcal{E})$ is the minimum number of dicliques such that each edge $e \in \mathcal{E}$ is contained in* exactly one *diclique.*

Tensors can be regarded as higher-order generalizations of vectors and matrices. In the following, we will only consider third-order tensors of the form $\mathbf{X} \in \mathbb{R}^{I \times J \times K}$, although many concepts generalize to higher-order tensors. The mode-$n$ unfolding (or matricization) of $\mathbf{X}$ arranges the mode-$n$ fibers of $\mathbf{X}$ as the columns of a newly formed matrix and will be denoted by $X_{(n)}$. The tensor-matrix product $\mathbf{A} = \mathbf{X} \times_n B$ multiplies the tensor $\mathbf{X}$ with the matrix $B$ along the $n$-th mode of $\mathbf{X}$ such that $A_{(k)} = BX_{(k)}$. For a detailed introduction to tensors and these operations we refer the reader to Kolda et al. [12]. The $k$-th frontal slice of a third-order tensor $\mathbf{X} \in \mathbb{R}^{I \times J \times K}$ will be denoted by $X_k \in \mathbb{R}^{I \times J}$. The outer product of vectors will be denoted by $\boldsymbol{a} \circ \boldsymbol{b}$. In contrast to matrices, there exist two non-equivalent notions of the rank of a tensor:

**Definition 4.** *Let $\mathbf{X} \in \mathbb{R}^{I \times J \times K}$ be a third-order tensor. The tensor rank $\mathrm{t\text{-}rank}(\mathbf{X})$ of $\mathbf{X}$ is defined as $\mathrm{t\text{-}rank}(\mathbf{X}) = \min \{r \mid \mathbf{X} = \sum_{i=1}^{r} \boldsymbol{a}_i \circ \boldsymbol{b}_i \circ \boldsymbol{c}_i\}$ where $\boldsymbol{a}_i \in \mathbb{R}^I$, $\boldsymbol{b}_i \in \mathbb{R}^J$, and $\boldsymbol{c}_i \in \mathbb{R}^K$. The multilinear rank $\mathrm{n\text{-}rank}(\mathbf{X})$ of $\mathbf{X}$ is defined as the tuple $(r_1, r_2, r_3)$, where $r_i = \mathrm{rank}\left(X_{(i)}\right)$.*

To model multi-relational data as tensors, we use the following concept of an adjacency tensor:

**Definition 5.** *Let $\mathcal{G} = \{(\mathcal{V}, \mathcal{E}_k)\}_{k=1}^{K}$ be a set of digraphs over the same set of vertices $\mathcal{V}$, where $|\mathcal{V}| = N$. The adjacency tensor of $\mathcal{G}$ is a third-order tensor $\mathbf{X} \in \{0,1\}^{N \times N \times K}$ with entries $x_{ijk} = 1$ if $v_i \rightsquigarrow v_j \in \mathcal{E}_k$ and $x_{ijk} = 0$ otherwise.*

For a single digraph, an adjacency tensor is equivalent to the digraph's adjacency matrix. Note that $K$ would correspond to the number of relation types in a domain.

## 2    On the Algebraic Complexity of Graph-Structured Data

In this section, we want to identify conditions under which tensor factorization can be considered efficient for relational learning. Let $\mathbf{X}$ denote an observed adjacency tensor with missing or noisy entries from which we seek to recover the true adjacency tensor $\mathbf{Y}$. Rank affects both the predictive as well as the runtime performance of a factorization: A high factorization rank will lead to poor runtime performance while a low factorization rank might not be sufficient to model $\mathbf{Y}$. We are therefore interested in identifying upper and lower bounds on the minimal rank — either tensor rank or multilinear rank — that is required such that a factorization can model the true adjacency tensor $\mathbf{Y}$. Please note that we are not concerned with bounds on the generalization error or the sample complexity that is needed to *learn* a good model, but on bounds on the algebraic complexity that is needed to *express* the true underlying data via factorizations. For sign-matrices $Y \in \{\pm 1\}^{N \times N}$, this question has been discussed in combinatorics and communication complexity via their *sign-rank* $\text{rank}_{\pm}(Y)$, which is the minimal rank needed to recover the sign-pattern of $Y$:

$$\text{rank}_{\pm}(Y) = \min_{M \in \mathbb{R}^{N \times N}} \left\{ \text{rank}(M) \mid \forall i, j : \text{sgn}(m_{ij}) = y_{ij} \right\}. \tag{1}$$

Although the concept of sign-rank can be extended to adjacency tensors, bounds based on the sign-rank would have only limited significance for our purpose, as no practical algorithms exist to find the solution to equation (1). Instead, we provide upper and lower bounds on tensor and multilinear rank, i.e. bounds on the *exact* recovery of $\mathbf{Y}$, for the following reasons: It follows immediately from (1) that any upper-bound on $\text{rank}(\mathbf{Y})$ will also hold for $\text{rank}_{\pm}(\mathbf{Y})$ since it has to hold that $\text{rank}_{\pm}(\mathbf{Y}) \leqslant \text{rank}(\mathbf{Y})$. Upper bounds on $\text{rank}(\mathbf{Y})$ can therefore provide insight under what conditions factorizations can be efficient on relational data — regardless whether we seek to recover exact values or sign patterns. Lower bounds on $\text{rank}(\mathbf{Y})$ provide insight under what conditions the exact recovery of $\mathbf{Y}$ can be inefficient. Furthermore, it can be observed empirically that lower bounds on the rank are more informative for existing factorization approaches to relational learning like [21, 13, 16] than bounds on sign-rank. For instance, let $S_n = 2I_n - J_n$ be the "signed identity matrix" of size $n$, where $I_n$ denotes the $n \times n$ identity matrix and $J_n$ denotes the $n \times n$ matrix of all ones. While it is known that $\text{rank}_{\pm}(S_n) = O(1)$ for *any* size $n$ [17], it can be checked empirically that SVD requires a rank larger than $\frac{n}{2}$, i.e. a rank of $O(n)$, to recover the sign pattern of $S_n$.

Based on these considerations, we state now the main theorem of this paper, which bounds the different notions of the rank of an adjacency tensor by the diclique partition number and the number of strongly connected components of the involved relations:

**Theorem 1.** *Tensor rank* $\text{t-rank}(\mathbf{Y})$ *and multilinear rank* $\text{n-rank}(\mathbf{Y}) = (r_1, r_2, r_3)$ *of any adjacency tensor* $\mathbf{Y} \in \{0,1\}^{N \times N \times K}$ *representing $K$ relations* $\{\Gamma_k(Y_k)\}_{k=1}^{K}$ *are bounded as*

$$\sum_{k=1}^{K} \text{dp}(\Gamma_k) \geqslant \theta \geqslant \max_k \text{scc}_+(\Gamma_k),$$

*where $\theta$ is any of the quantities* $\text{t-rank}(\mathbf{Y})$, $r_1$, *or* $r_2$.

To prove theorem 1 we will first derive upper and lower bounds on adjacency *matrices* and then show how these bounds generalize to adjacency *tensors*.

**Lemma 1.** *For any adjacency matrix* $Y \in \{0,1\}^{N \times N}$ *it holds that* $\text{dp}(\Gamma) \geqslant \text{rank}(Y) \geqslant \text{scc}_+(\Gamma)$.

*Proof.* The upper bound of lemma 1 follows directly from the fact that $\text{dp}(\Gamma(Y)) = \text{rank}_{\mathbb{N}}(Y)$ and the fact that $\text{rank}_{\mathbb{N}}(Y) \geqslant \text{rank}(Y)$, where $\text{rank}_{\mathbb{N}}(Y)$ denotes the *non-negative integer rank* of the binary matrix $Y$ [19, see eq. 1.6.5 and eq. 1.7.1]. □

Next we will prove the lower bound of lemma 1. Let $\lambda_i(Y)$ denote the $i$-th (complex) *eigenvalue* of $Y$ and let $\Lambda(Y)$ denote the *spectrum* of $Y \in \mathbb{R}^{N \times N}$, i.e. the multiset of (complex) eigenvalues of $Y$. Furthermore, let $\rho(Y) = \max_i |\lambda_i(Y)|$ be the *spectral radius* of $Y$. Now, recall the celebrated Perron-Frobenius theorem:

**Theorem 2** ([25, Theorem 8.2]). *Let* $Y \in \mathbb{R}^{N \times N}$ *with* $y_{ij} \geqslant 0$ *be a non-negative irreducible matrix. Then* $\rho(Y) > 0$ *is a simple eigenvalue of $Y$ associated with a positive eigenvector.*

Please note that a nontrivial digraph is strongly connected iff its adjacency matrix is irreducible [3, Theorem 3.2.1]. Furthermore, an adjacency matrix is nilpotent iff the associated digraph is acyclic [3, Section 9.8]. Hence, the adjacency matrix of a strongly connected component $\Psi$ is nilpotent iff $\Psi$ is trivial. Given these considerations, we can now prove the lower bound of lemma 1:

**Lemma 2.** *For any non-negative adjacency matrix $Y \in \mathbb{R}^{N \times N}$ with $y_{ij} \geqslant 0$ of a weighted digraph $\Gamma$ it holds that* $\mathrm{rank}(Y) \geqslant \mathrm{scc}_+(\Gamma)$.

*Proof.* Let $\Gamma$ consist of $k$ nontrivial strongly connected components. The Frobenius normal form $B$ of its associated adjacency matrix $Y$ consists then of $k$ irreducible matrices $B_i$ on its block diagonal. It follows from theorem 2 that each irreducible $B_i$ has at least one nonzero eigenvalue. Since $B$ is block upper triangular, it holds also that $\Lambda(B) = \bigcup_{i=1}^{k} \Lambda(B_i)$. As the rank of a square matrix is larger or equal to the number of its nonzero eigenvalues, it follows that $\mathrm{rank}(B) \geqslant k$. Lemma 2 follows from the fact that $B$ is similar to $Y$ and that matrix similarity preserves rank. $\square$

So far, we have shown that $\mathrm{rank}(Y)$ of an adjacency *matrix Y* is bounded by the diclique covering number and the number of nontrivial strongly connected components of the associated digraph. To complete the proof of theorem 1 we will now show that these bounds for unirelational data translate directly to multi-relational data and to the different notions of the rank of an adjacency *tensor*. In particular we will show that both notions of tensor rank are lower bounded by the maximum rank of a single frontal slice in the tensor and upper bounded by the sum of the ranks of all frontal slices:

**Lemma 3.** *The tensor rank* t-rank$(\mathbf{Y})$ *and multilinear rank* n-rank$(\mathbf{Y}) = (r_1, r_2, r_3)$ *of any third-order tensor* $\mathbf{Y} \in \mathbb{R}^{I \times J \times K}$ *with frontal slices $Y_k$ are bounded as*

$$\sum_{k=1}^{K} \mathrm{rank}(Y_k) \geqslant \theta \geqslant \max_k \mathrm{rank}(Y_k),$$

*where $\theta$ is any of the quantities* t-rank$(\mathbf{Y})$, $r_1$, *or* $r_2$.

*Proof.* Due to space constraints, we will include only the proof for tensor rank. The proof for multilinear rank can be found in supplementary material A.1. Let t-rank$(\mathbf{Y}) = r$ and $\mathrm{rank}(Y_k) = r_{\max}$. It can be seen from the definition of tensor rank that $Y_k = \sum_{i=1}^{r} c_{kr}(\boldsymbol{a}_r \boldsymbol{b}_r^\top)$. Consequently, it follows from the subadditivity of matrix rank, i.e. $\mathrm{rank}(A + B) \leqslant \mathrm{rank}(A) + \mathrm{rank}(B)$, that

$$r_{\max} = \mathrm{rank}\left(\sum_{i=1}^{r} c_{kr} \boldsymbol{a}_r \boldsymbol{b}_r^\top\right) \leqslant \sum_{i=1}^{r} \mathrm{rank}\left(c_{kr} \boldsymbol{a}_r \boldsymbol{b}_r^\top\right) \leqslant r$$

where the last inequality follows from $\mathrm{rank}\left(c_{kr} \boldsymbol{a}_r \boldsymbol{b}_r^\top\right) \leqslant 1$. Now we will derive the upper bound of lemma 3 by providing a decomposition of $\mathbf{Y}$ with rank $r = \sum_k \mathrm{rank}(Y_k)$ that recovers $\mathbf{Y}$ exactly. Let $Y_k = U_k S_k V_k^\top$ be the SVD of $Y_k$ with $S_k = \mathrm{diag}(\boldsymbol{s}_k)$. Furthermore, let $U = [U_1 \, U_2 \, \cdots \, U_K]$, $V = [V_1 \, V_2 \, \cdots \, V_K]$, and let $S$ be a block-diagonal matrix where the $i$-th block on the diagonal is equal to $\boldsymbol{s}_i^\top$ and all other entries are 0. It can be easily verified that $\sum_{i=1}^{r} \hat{\boldsymbol{u}}_i \circ \hat{\boldsymbol{v}}_i \circ \hat{\boldsymbol{s}}_i$ provides an exact decomposition of $\mathbf{Y}$, where $r = \sum_k \mathrm{rank}(Y_k)$ and $\hat{\boldsymbol{u}}_i$, $\hat{\boldsymbol{v}}_i$, and $\hat{\boldsymbol{s}}_i$ are the $i$-th columns of the matrices $U$, $V$, and $S$. The inequality in lemma 3 follows since $r$ is not necessarily minimal. $\square$

Theorem 1 can now be derived by combining lemmas 1 and 3 what concludes the proof.

**Discussion** It can be seen from theorem 1 that factorizations can be computationally efficient when $\sum_k \mathrm{dp}(\Gamma_k)$ is small. However, factorizations can potentially be inefficient when $\mathrm{scc}_+(\Gamma_k)$ is large for *any* $\Gamma_k$ in the data. For instance, consider an idealized *marriedTo* relation, where each person is married to exactly one person. Evidently, for $m$ marriages, the associated digraph would consist of $m$ strongly connected components, i.e. one component for each marriage. According to lemma 2, a factorization model would at least require $m$ latent components to recover this adjacency matrix exactly. Consequently, an algorithm with cubic runtime complexity in the rank would only be able to recover $\mathbf{Y}$ for this relation when the number of marriages is small, what limits its applicability to these relations. A second important observation for multi-relational learning is that the lower bound in theorem 1 depends only on the largest rank of a single frontal slice (i.e. a single adjacency matrix) in $\mathbf{Y}$. For multi-relational learning this means that regularities between different relations can not decrease tensor or multilinear rank below the largest matrix rank of a single relation. For instance, consider an $N \times N \times 2$ tensor $\mathbf{Y}$ where $Y_1 = Y_2$. Clearly it holds that $\mathrm{rank}(Y_{(3)}) = 1$, such that $Y_1$ could easily be predicted from $Y_2$ when $Y_2$ is known. However, theorem 1 states that the rank of the factorization must be at least $\mathrm{rank}(Y_1)$ — which can be arbitrarily large up to $N$ — when

the first two modes of $\mathbf{Y}$ are also factorized. Please note that this is not a statement about sample complexity or generalization error which *can be* reduced when factorizing all modes of a tensor, but a statement about the minimal rank that is required to *express* the data. A last observation from the previous discussion is that factorizations and observable variable methods excel at different aspects of relationship prediction. For instance, predicting relationships in the idealized *marriedTo* relation can be done easily with Horn clauses and link predication heuristics as listed in supplementary material A.2. In contrast, factorization methods would be inefficient in predicting links in this relation as they would require at least one latent component for each marriage. At the same time, links in a diclique of any size can trivially be modeled with a rank-2 factorization that indicates the partition memberships, while standard neighborhood-based methods will fail on dicliques since — by the definition of a diclique — there do not exist links within one partition yet the only vertices that share neighbors are located in the same partition.

## 3  An Additive Relational Effects Model

RESCAL is a state-of-the-art relational learning method that is based on a constrained Tucker-decomposition and as such is subject to bounds as in theorem 1. Motivated by the results of section 2, we propose an additive tensor decomposition approach to combine the strengths of latent and observable variable methods to reduce the rank requirements of RESCAL on multi-relational data. To include the information of observable pattern methods in the factorization, we augment the RESCAL model with an additive term that holds the predictions of observable pattern methods. In particular, let $\mathbf{X} \in \{0,1\}^{N \times N \times K}$ be a third-order adjacency tensor and $\mathbf{M} \in \mathbb{R}^{N \times N \times P}$ be a third-order tensor that holds the predictions of an arbitrary number of relational learning methods. The proposed *additive relational effects* model (ARE) decomposes $\mathbf{X}$ into

$$\mathbf{X} \approx \mathbf{R} \times_1 A \times_2 A + \mathbf{M} \times_3 W, \tag{2}$$

where $A \in \mathbb{R}^{N \times r}$, $\mathbf{R} \in \mathbb{R}^{r \times r \times K}$ and $W \in \mathbb{R}^{K \times P}$. The first term of equation (2) corresponds to the RESCAL model which can be interpreted as following: The matrix $A$ holds the latent variable representations of the entities, while each frontal slice $R_k$ of $\mathbf{R}$ is an asymmetric $r \times r$ matrix that models the interactions of the latent components for the $k$-th relation. The variable $r$ denotes the number of latent components of the factorization. An important aspect of RESCAL for relational learning is that entities have a unique latent representation via the matrix $A$. This enables a relational learning effect via the propagation of information over different relations and the occurrences of entities as a subject or objects in relationships. For a detailed description of RESCAL we refer the reader to Nickel et al. [21, 22]. After computing the factorization (2), the score for the existence of a single relationship is calculated in ARE via $\hat{x}_{ijk} = \boldsymbol{a}_i^T R_k \boldsymbol{a}_j + \sum_{p=1}^P w_{kp} m_{ijp}$.

The construction of the tensor $\mathbf{M}$ is of the following: Let $\mathcal{F} = \{f_p\}_{p=1}^P$ be a set of given real-valued functions $f_p : \mathcal{V} \times \mathcal{V} \rightarrow \mathbb{R}$ which assign scores to each pair of entities in $\mathcal{V}$. Examples of such score functions include link prediction heuristics such as Common Neighbors, Katz Centrality, or Horn clauses. Depending on the underlying model these scores can be interpreted as confidences value or as probabilities that a relationship exists between two entities. We collect these real-valued predictions of $P$ score functions in the tensor $\mathbf{M} \in \mathbb{R}^{N \times N \times P}$ by setting $m_{ijp} = f_p(v_i, v_j)$. Supplementary material A.2 provides a detailed description of the construction of $\mathbf{M}$ for typical score functions. The tensor $\mathbf{M}$ acts in the factorization as an independent source of information that predicts the existence of relationships. The term $\mathbf{M} \times_3 W$ can be interpreted as learning a set of weights $w_{kp}$ which indicate how much the $p$-th score function in $\mathbf{M}$ correlates with the $k$-th relation in $\mathbf{X}$. For this reason we refer to $\mathbf{M}$ also as the *oracle tensor*. If $\mathbf{M}$ is composed of relation path features as proposed by Lao et al. [15], the term $\mathbf{M}W$ is closely related to the Path Ranking Algorithm (PRA) [15].

The main idea of equation (2) is the following: The term $\mathbf{R} \times_1 A \times_2 A$ is equivalent to the RESCAL model and provides an efficient approach to learn from latent patterns on relational data. The oracle tensor $\mathbf{M}$ on the other hand is not factorized, such that it can hold information that is difficult to predict via latent variable methods. As it is not clear a priori which score functions are good predictors for which relations, the term $\mathbf{M} \times_3 W$ learns a weighting of how predictive any score function is for any relation. By integrating both terms in an additive model, the term $\mathbf{M} \times_3 W$ can potentially reduce the required rank for the RESCAL term by explaining links that, for instance, reduce the diclique partition number of a digraph. Rules and operations that are likely to reduce the diclique partition

number of slices in $\mathbf{X}$ are therefore good candidates to be included in $\mathbf{M}$. For instance, by including a copy of the observed adjacency tensor $\mathbf{X}$ in $\mathbf{M}$ (or some selected frontal slices $X_k$), the term $\mathbf{M} \times_3 W$ can easily model common multi-relational patterns where the existence of a relationship in one relation correlates with the existence of a relationship between the same entities in another relation via $x_{ijk} = \sum_{p \neq k} w_{kp} x_{ijp}$. Since $w_{kp}$ is allowed to be negative, anti-correlations can be modeled efficiently. ARE is similar in spirit to the model of Koren [14], which extends SVD with additive terms to include local neighborhood information in an uni-relational recommendation setting and Jiang et al. [9] which uses an additive matrix factorization model for link prediction. Furthermore, the recently proposed Google Knowledge Vault (KV) [5] considers a combination of PRA and a neural network model related to RESCAL for learning from large multi-relational datasets. However, in KV both models are trained separately and combined only later in a separate fusion step, whereas ARE learns both models jointly what leads to the desired rank-reduction effect.

To compute ARE, we pursue a similar optimization scheme as used for RESCAL which has been shown to scale to large datasets [22]. In particular, we solve the regularized optimization problem

$$\min_{A, \mathbf{R}, W} \|\mathbf{X} - (\mathbf{R} \times_1 A \times_2 A + \mathbf{M} \times_3 W)\|_F^2 + \lambda_A \|A\|_F^2 + \lambda_R \|\mathbf{R}\|_F^2 + \lambda_W \|W\|_F^2. \qquad (3)$$

via alternating least-squares, which is a block-coordinate optimization method in which blocks of variables are updated alternatingly until convergence. For equation (3) the variable blocks are given naturally by the factors $A$, $\mathbf{R}$, and $W$.

**Updates for $W$**  Let $\mathbf{E} = (\mathbf{X} - \mathbf{R} \times_1 A \times_2 A)$ and $I$ be the identity matrix. We rewrite equation (2) as $E_{(3)} \approx W M_{(3)}$ such that equation (3) becomes a regularized least-squares problem when solving for $W$. It follows that updates for $W$ can be computed via $W \leftarrow (M_{(3)} M_{(3)}^\top + \lambda_W I)^{-1} M_{(3)} E_{(3)}^\top$. However, performing the updates in this way would be very inefficient as it involves the computation of the *dense $N \times N \times K$ tensor* $\mathbf{R} \times_1 A \times_2 A$. This would quickly lead to scalability issues with regard to runtime and memory requirements. To overcome this issue, we rewrite $M_{(3)} E_{(3)}^\top$ using the equality $(\mathbf{R} \times_1 A \times_2 A)_{(3)} M_{(3)}^\top = R_{(3)} (\mathbf{M} \times_1 A^\top \times_2 A^\top)_{(3)}^\top$. Updates for $W$ can then be computed efficiently as

$$W^\top \leftarrow \left[ X_{(3)} M_{(3)}^\top - R_{(3)} (\mathbf{M} \times_1 A^\top \times_2 A^\top)_{(3)}^\top \right] (M_{(3)} M_{(3)}^\top + \lambda_W I)^{-1}. \qquad (4)$$

In equation (4) the dense tensor $\mathbf{R} \times_1 A \times_2 A$ is never computed explicitly and the computational complexity with regard to the parameters $N$, $K$, and $r$ is reduced from $O(N^2 K r)$ to $O(N K r^3)$. Furthermore, all terms in equation (4) except $R_{(3)} (\mathbf{M} \times_1 A^\top \times_2 A^\top)_{(3)}^\top$ are constant and have only to be computed once at the beginning of the algorithm. Finally, $X_{(3)} M_{(3)}^\top$ and $M_{(3)} M_{(3)}^\top$ are the products of *sparse* matrices such that their computational complexity depends only on the number of nonzeros in $\mathbf{X}$ or $\mathbf{M}$. A full derivation of equation (4) can be found in the supplementary material A.4.

**Updates for $A$ and $\mathbf{R}$**  The updates for $A$ and $\mathbf{R}$ can be derived directly from the RESCAL-ALS algorithm by setting $\mathbf{E} = \mathbf{X} - \mathbf{M} \times_3 W$ and computing the RESCAL factorization of $\mathbf{E}$. The updates for $A$ can therefore be computed by:

$$A \leftarrow \left( \sum_{k=1}^{K} E_k A R_k^\top + E_k^\top A R_k \right) \left( \sum_{k=1}^{K} R_k A^\top A R_k^\top + R_k^\top A^\top A R_k + \lambda I \right)^{-1}$$

where $E_k = X_k - \mathbf{M} \times_3 w_k$ and $w_k$ denotes the $k$-th row of $W$.

The updates of $\mathbf{R}$ can be computed in the following way: Let $A = U \Sigma V^\top$ be the SVD of $A$, where $\sigma_i$ is the $i$-th singular value of $A$. Furthemore, let $S$ be a matrix with entries $s_{ij} = \sigma_i \sigma_j / (\sigma_i^2 \sigma_j^2 + \lambda_R)$. An update of $R_k$ can then be computed via $R_k \leftarrow V \left( S * (U^\top (X_k - \mathbf{M} \times_3 w_k) U) \right) V^\top$, where "$*$" denotes the Hadamard product. For a full derivation of these updates please see [20].

## 4  Evaluation

We evaluated ARE on various multi-relational datasets where we were in particular interested in its generalization ability relative to the factorization rank. For comparison, we included the well-known

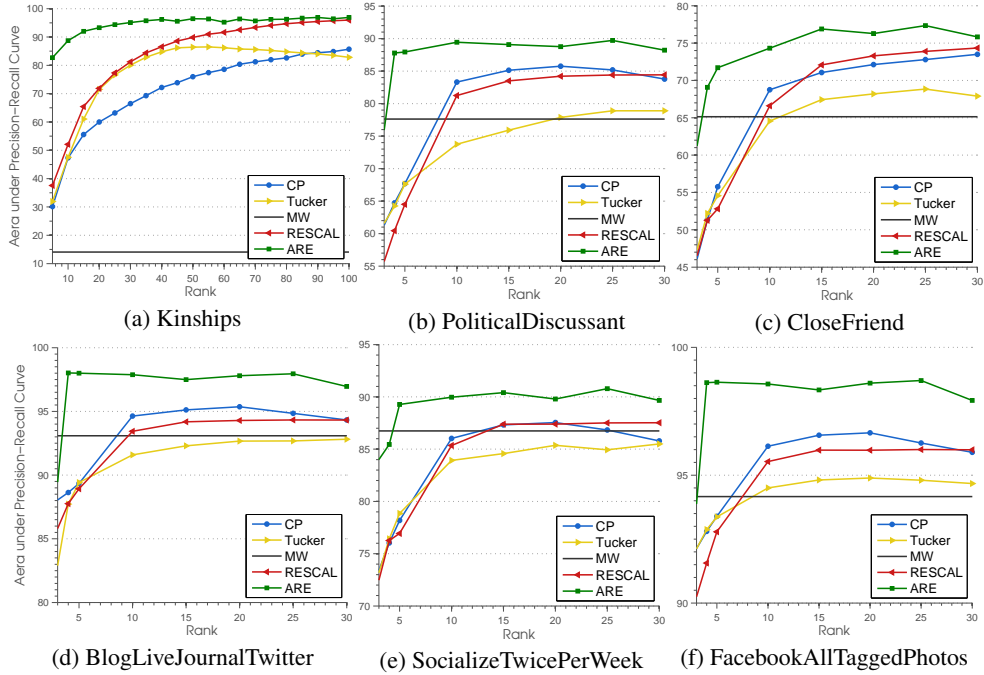

Figure 1: Evaluation results for AUC-PR on the Kinships (1a) and Social Evolution data sets (1b-1f).

CP and Tucker tensor factorizations in the evaluation, as well as RESCAL and the non-latent model $\mathbf{X} \approx \mathbf{M} \times_3 W$ (in the following denoted by $\mathbf{M}W$). In all experiments, the oracle tensor $\mathbf{M}$ used in $\mathbf{M}W$ and ARE is identical, such that the results of $\mathbf{M}W$ can be regarded as a baseline for the contribution of the heuristic methods to ARE. Following [10, 11, 28, 21] we used k-fold cross-validation for the evaluation, partitioning the entries of the adjacency tensor into training, validation, and test sets. In the test and validation folds all entries are set to 0. Due to the large imbalance of true and false relationships, we used the area under the precision-recall curve (AUC-PR) to measure predictive performance, which is known to behave better with imbalanced classes then AUC-ROC. All AUC-PR results are averaged over the different test-folds. Links and references for the datasets used in the evaluation are provided in the supplementary material A.5.

**Social Evolution** First, we evaluated ARE on a dataset consisting of multiple relations of persons living in an undergraduate dormitory. From the relational data, we constructed a $84 \times 84 \times 5$ adjacency tensor where two modes correspond to persons and the third mode represents the relations between these persons such as friendship (*CloseFriend*), social media interaction (*BlogLivejournalTwitter* and *FacebookAllTaggedPhotos*), political discussion (*PoliticalDiscussant*), and social interaction (*SocializeTwicePerWeek*). For each relation, we performed link prediction via 5-fold cross validation. The oracle tensor $\mathbf{M}$ consisted only of a copy of the observed tensor $\mathbf{X}$. Including $\mathbf{X}$ in $\mathbf{M}$ allows ARE to efficiently exploit patterns where the existence of a social relationship for a particular pair of persons is predictive for other social interactions between exactly this pair of persons (e.g. close friends are more likely to socialize twice per week). It can be seen from the results in figure $1(b-f)$ that ARE achieves better performance than all competing approaches and already achieves excellent performance at a very low rank, what supports our theoretical considerations.

**Kinship** The Kinship dataset describes the kinship relations in the Australian Alyawarra tribe in terms of 26 kinship relations between 104 persons. The task in the experiment was to predict unknown kinship relations via 10-fold cross validation in the same manner as in [21]. Table 1 shows the improvement of ARE over state-of-the-art relational learning methods. Figure 1a shows the predictive performance compared to the rank of multiple factorization methods. It can be seen that ARE outperforms all other methods significantly for lower rank. Moreover, starting from rank 40 ARE gives already comparable results to the best results in table 1. As in the previous experiments, $\mathbf{M}$ consisted only of a copy of $\mathbf{X}$. On this dataset, the copy of $\mathbf{X}$ allows ARE to model efficiently that the relations in the data are mutually exclusive by setting $w_{ii} > 0$ and $w_{ij} < 0$ for all $i \neq j$. This also explains the large improvement of ARE over RESCAL for small ranks.

**Link Prediction on Semantic Web Data**  The SWRC ontology models a research group in terms of people, publications, projects, and research interests. The task in our experiments was to predict the affiliation relation, i.e. to map persons to research groups. We followed the experimental setting in [18]: From the raw data, we created a $12058 \times 12058 \times 85$ tensor by considering all directly connected entities of persons and research groups. In total, 168 persons and 5 research groups are considered in the evaluation data. The oracle tensor $\mathbf{M}$ consisted again of a copy of $\mathbf{X}$ and of the common neighbor heuristics $X_i X_i$ and $X_i^\top X_i^\top$. These heuristics were included to model patterns like *people who share the same research interest are likely in the same affiliation* or *a person is related to a department if the person belongs to a group in the department*. We also imposed a sparsity penalty on $W$ to prune away inactive heuristics during iterations. Table 2 shows that ARE improved the results significantly over three state-of-the-art link prediction methods for Semantic Web data. Moreover, whereas RESCAL required a rank of 45, ARE required only a small rank of 15.

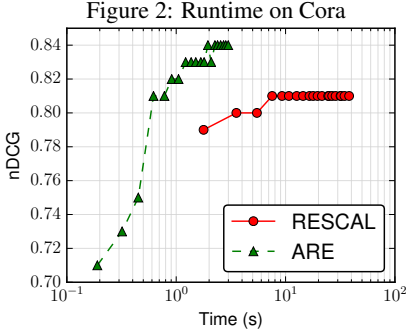

Figure 2: Runtime on Cora

Table 1: Evaluation Results on Kinships.

|  | MRC [11] | BCTF [28] | LFM [8] | RESCAL | ARE |
|---|---|---|---|---|---|
| AUC | 86 | 90 | 94.6 | 96 | **96.9** |
| Rank | - | - | (50,50,500) | 100 | **90** |

Table 2: Evaluation results on SWRC.

|  | SVD | Subtrees [18] | RESCAL | MW | ARE |
|---|---|---|---|---|---|
| nDCG | 0.8 | 0.95 | 0.96 | 0.59 | **0.99** |

**Runtime Performance**  To evaluate the trade-off between runtime and predictive performance we recorded the nDCG values of RESCAL and ARE after each iteration of the respective ALS algorithms on the Cora citation database. We used the variant of Cora in which all publications are organized in a hierarchy of topics with two to three levels and 68 leaves. The relational data consists of information about paper citations, authors and topics from which a tensor of size $28073 \times 28073 \times 3$ is constructed. The oracle tensor consisted of a copy of $\mathbf{X}$ and the common neighbor patterns $X_i X_j$ and $X_i^\top X_j^\top$ to model patterns such that a cited paper shares the same topic, a cited paper shares the same author etc. The task of the experiment was to predict the leaf topic of papers by 5-fold cross-validation on a moderate PC with Intel(R) Core i5 @3.1GHz, 4G RAM. The optimal rank 220 for RESCAL was determined out of the range $[10, 300]$ via parameter selection. For ARE we used a significantly smaller rank 20. Figure 2 shows the runtime of RESCAL and ARE compared to their predictive performance. It is evident that ARE outperforms RESCAL after a few iterations although the rank of the factorization is decreased by an order of magnitude. Moreover, ARE surpasses the best prediction results of RESCAL in terms of total runtime even before the first iteration of RESCAL-ALS has terminated.

## 5   Concluding Remarks

In this paper we considered learning from latent and observable patterns on multi-relational data. We showed analytically that the rank of adjacency tensors is upper bounded by the sum of diclique partition numbers and lower bounded by the maximum number of strongly connected components of any relation in the data. Based on our theoretical results, we proposed an additive tensor factorization approach for learning from multi-relational data which combines strengths from latent and observable variable methods. Furthermore we presented an efficient and scalable algorithm to compute the factorization. Experimentally we showed that the proposed approach does not only increase the predictive performance but is also very successful in reducing the required rank — and therefore also the required runtime — of the factorization. The proposed additive model is one option to overcome the rank-scalability problem outlined in section 2, however not the only one. In future work we intend to investigate to what extent sparse or hierarchical models can be used to the same effect.

**Acknowledgements**  Maximilian Nickel acknowledges support by the Center for Brains, Minds and Machines (CBMM), funded by NSF STC award CCF-1231216. We thank Youssef Mroueh and Lorenzo Rosasco for clarifying discussions on the theoretical part of this paper.

## Footnotes

[1]Similar results can be obtained for state-of-the-art algorithms to compute the well-known CP and Tucker decompositions. Please see the supplementary material A.3 for the respective derivations.

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
