[Supplementary Material]

# A Supplementary Material

## A.1 Proof for Upper Bound on Multilinear Rank

In the following we will proof the upper bound on multilinear rank for lemma 3

*Proof.* Let $\text{rank}(X_m) = r_{\max}$. It follows from the basic properties of matrix rank that $X_m$ has $r_{\max}$ linearly independent rows. Since the unfolding of $\mathbf{X}$ in the first and second mode is a block matrix where the $k$-th block corresponds to $X_k$ or its transpose, i.e.

$$X_{(1)} = \begin{bmatrix} X_1 & X_2 & \ldots & X_K \end{bmatrix}$$
$$X_{(2)} = \begin{bmatrix} X_1^\top & X_2^\top & \ldots & X_K^\top \end{bmatrix},$$

it follows that $X_{(1)}$ and $X_{(2)}$ have also at least $r_{\max}$ linearly independent rows and at most $\sum_k \text{rank}(X_k)$ independent rows, such that $\sum_k \text{rank}(X_k) \geqslant r_1, r_2 \geqslant r_{\max}$ □

## A.2 Link Prediction Methods

Table 3 lists typical examples for relational learning functions that we will consider for the construction of $\mathbf{M}$. In table 3 $\text{score}(v_1, v_2)$ denotes the score that a function assigns to a link, while $N(v_1)$ denotes

Table 3: Link Prediction Heuristics

| Method | $\text{score}(v_1, v_2)$ |
|---|---|
| Common Neighbors | $\|N(v_1) \cap N(v_2)\|$ |
| Jaccard Coefficient | $\dfrac{\|N(v_1) \cap N(v_2)\|}{\|N(v_1) \cup N(v_2)\|}$ |
| Adamic/Adar | $\sum_{z \in N(v_1) \cap N(v_2)} (\log \|N(z)\|)^{-1}$ |
| Katz | $\sum_k^\infty \beta^k \|\text{paths}(v_1, v_2, k)\|$ |
| Horn Clause | $\begin{cases} 1, & \text{if } P_1 \wedge P_2 \wedge \ldots \wedge P_n \\ 0, & \text{else.} \end{cases}$ |

the set of neighbors of vertex $v_1$, and $\text{path}(v_1, v_2, k)$ denotes the set of all paths between vertices $v_1$ and $v_2$ of length $k$. For the definition of neighborhood in a digraph see definition 6:

**Definition 6** (Neighorhood). *Let $\Gamma = (\mathcal{V}, \mathcal{E})$ be a digraph. The* in-neighborhood *of a vertex $v$ is defined as $N^-(v) = \{u | u \rightsquigarrow v \in \mathcal{E}\}$, the* out-neighborhood *is defined as $N^+(v) = \{u | v \rightsquigarrow u \in \mathcal{E}\}$, and the* neighborhood *of $v$ is defined as $N^-(v) \cup N^+(v)$.*

## A.3 Computational Complexity of Tensor Factorizations

Here, we review the computational complexity of standard algorithms to compute tensor factorizations with regard to the rank of an adjacency tensor. Alternating least-squares algorithms (ALS) are the "workhorse" algorithms to compute the CP decomposition and the RESCAL factorization, while the higher-order orthogonal iterations (HOOI) algorithm is commonly used to computed the Tucker decomposition [2, 6].

Figure 3: Illustration of a diclique.

It can be verified easily that one iteration of CP-ALS scales quadratic with the number of entities and cubic with the number of latent components $r$ when factorizing an adjacency tensor $\mathbf{X} \in \{0,1\}^{N \times N \times K}$. To compute a single update of $C$ (and analogously for $A$ and $B$), the following term has to be computed

$$C \leftarrow X_{(1)}(B \odot A)(B^\top B * A^\top A)^\dagger. \tag{5}$$

Since $B^\top B * A^\top A$ is a $r \times r$ matrix, the computational complexity of $(B^\top B * A^\top A)^\dagger$ is $O(r^3)$. Furthermore, $B \odot A$ is an $N^2 \times r$ matrix, such that its computation needs $O(N^2 r)$ operations.

To compute the Tucker decomposition using HOOI, it is necessary to compute the $r_1$ largest eigenvectors of $Y_{(1)} Y_{(1)}^\top$ where $Y_{(1)} = G_{(1)}(C \otimes B)^\top$. Since $Y_{(1)}$ is an $N \times r_2 r_3$ matrix, the matrix product $Y_{(1)} Y_{(1)}^\top$ alone already needs $O(N^2 r_2 r_3)$ operations. Furthermore, to compute the $r_1$ largest eigenvectors of a $N \times N$ matrix the implicitly restarted Arnoldi method (IRAM) is used which has a computational complexity of $O(Nr_1^2)$ [3]. Derivations for $r_2$ and $r_3$ are analogous.

RESCAL-ALS scales linearly with the data size, i.e. linearly with the number of entities, number of relations, and the number of known facts. The computational complexity with regard to the number of latent components $r$ is $O(r^3)$. For a full derivation of the runtime complexity see Nickel [5].

### A.4 Derivation of Updates for $R_k$ and $W$

The improve updates for $W$ can be derive from the following equality

$$
\begin{aligned}
(R \times_1 A \times_2 A)_{(3)} M_{(3)}^\top &= R_{(3)}(A \otimes A)^\top M_{(3)}^\top \\
&= R_{(3)} \left( M_{(3)}(A \otimes A) \right)^\top \\
&= R_{(3)} \left( \mathbf{M} \times_1 A^\top \times_2 A^\top \right)_{(3)}^\top
\end{aligned}
$$

Please note that $A$ is not required to be orthonormal.

The runtime complexity of computing $(R \times_1 A \times_2 A)_{(3)} M_{(3)}^\top$ is $O(N^2 Kr + \mathrm{nnz}(M)N)$, while the computational complexity of $R_{(3)}(\mathbf{M} \times_1 A^\top \times_2 A^\top)_{(3)}$ is only $O(NKr^3 + \mathrm{nnz}(M)r)$.

### A.5 Datasets

The datasets used in the evaluation are available from the following locations:

| | |
|---|---|
| Social Evolution [4] | `http://realitycommons.media.mit.edu/socialevolution.html` |
| Kinships Denham [1] | `http://alchemy.cs.washington.edu/data/kinships/` |
| SWRC [7] | `http://ontoware.org/swrc/` |
| Cora | `https://people.cs.umass.edu/~mccallum/data.html` |