[Reviews · NeurIPS 2014]

Submitted by Assigned_Reviewer_19

The paper develops lower and upper bounds on
the required rank of adjacency tensor factorizations
to recover the adjacency tensor. This is, it investigates
the problem of the minimal rank required to express
the true underlying data via factorizations.
These bounds are shown to of practical use by scaling RESCAL
up.

The paper is extremely well written and makes several
interesting and important contributions. Moreover,
the main theoretical result presented in Theorem 1 is quite
intuitive and (as much as an informed outsider can say) novel.
The proofs (as far as checked) are also correct. Moreover, the
modified RESCAL version, called ARE, is well developed and shown to
perform and scale better than RESCAL on a number of datasets.

To summarize, I do not have any negative comments. Well done.

The authors might also be interested in the rather recent paper

Bhaskara, Charikar, Vijayaraghavan
Uniqueness of Tensor Decompositions and Identifiability
in Latent Variable Models
COLT 2014

to see whether the results can actually be transferred to
other latent variable models.
Summary: The paper is extremely well written and makes several
interesting and important contributions. Moreover, the
modified RESCAL version, called ARE, is well developed and shown to
perform and scale better than RESCAL on a number of datasets.

Submitted by Assigned_Reviewer_31

The factorization rank which is also the number of latent variables is an important parameter in using matrix factorization methods. It affects both model predictive performance and runtime. Most importantly the scalability of most implemented methods for matrix factorization is limited by this parameter.

This paper derives upper bound (number of strongly connected components) and lower bound (diclique partition number) on the required rank to recover adjacency tensors and determine the conditions under which factorization is an efficient approach for learning from relational data. The paper provides analysis on tensor rank and multi-linear rank.

They provide three interesting insights from their analysis about efficiency of factorization methods with respect to the number of non-trivial connected components in relations, independence of rank and regularities between relations, and relation of dicliques with efficiency of factorization methods in comparison with common similarity metrics.
Based on these findings an additive factorization model called additive relational effects model (ARE) is proposed. ARE augments RESCAL model with an additive term that holds the predictions of observable pattern methods.

While the paper provides interesting insights and theoretical analysis seems valid, the use of oracle tensor in the evaluation is not very clear; When the ARE is defined the oracle tensor is formed by putting in predictions from heuristic similarity measures such as Jaccard or common neighbors. In the empirical evaluation however it is mentioned that a copy of observed tensor is used as M.
Yet, it is not clear if any of the similarity heuristics were used in construction of the oracle tensor in the evaluation section. If they were used, does the other compared methods perform factorization on the oracle tensor or the original observed one? In the former case, the performance boost of ARE could be due to benefiting mostly from the information of similarity heuristics, which is not a fair comparison. If they are not used, some performance analysis of including these measures in the oracle tensor could be discussed. The paper provides related arguments of this matter in section 3, and it would be beneficial to clearly discuss the effect of including them in the evaluation section.
Summary: The paper provide bounds on factorization rank, an discusses interesting insights and theoretical analysis. It then proposes an additive factorization model called ARE. Some clarifications and discussions about using similarity heuristics in the oracle tensor is needed to make the paper easier to follow in the discussion section.

Submitted by Assigned_Reviewer_41

The paper provides a 3D tensor approach to link discovery in relational domains represented by labeled directed graphs. The paper is clearly written and understandable, with the following caveat. It assumes some familiarity with tensors; it would be nice if it could review the necessary material, but this is probably infeasible given the space constraints.

The primary contribution of the paper is to provide upper and lower bounds on adjacency tensor rank, and then to use these to construct an alternative algorithm for tensor-based link discovery. To my knowledge the bounds and algorithm are novel, and the strong empirical results across a range of tasks convince me of potential significance. An added point of significance is that the incorporation of the "oracle" tensor M makes it possible to easily bring more background knowledge into the algorithm if available. Nevertheless the authors make the right decision not to bring in such background knowledge for most of the empirical results, so that the comparison with other approaches is a fair one.
Summary: The paper proves upper and lower bounds on adjacency tensor rank and, based on these, presents a new algorithm for link discovery in labeled directed graphs that gives strong empirical results over multiple relational learning benchmark tasks.
Author Feedback
Author rebuttal: We thank all reviewers for their thoughtful and helpful reviews.

Reviewer 1:
Thank you for your comments and the very interesting reference. We were aware of the related paper [Anandkumar et al, 2012] and will take a close look to the paper you suggested. We agree that it would be interesting to see if a similar form of our argumentation could be applied to the setting considered in these papers.

Reviewer 2:
Thank you for your comments. We agree that it is important to provide a fair comparison to factorization methods without heuristics as well as insights into the benefits of adding heuristics.
We believe that the experiments described in the paper provide both aspects:
For Social Evolution and Kinships -- where we compared ARE to other tensor factorizations -- the oracle tensor M consisted *only* of a copy of X and therefore did not incorporate any heuristic information. As such it provides a fair comparison to pure factorization methods. For SWRC and Cora, the oracle tensor consisted of a copy of X plus heuristic common neighbour patterns. The purpose of the Cora experiments was to measure how beneficial the addition of heuristic information is when incorporated in ARE over the pure latent variable method RESCAL in terms of runtime. On SWRC we compared ARE for instance to [16] which also uses (considerably more complex) neighbourhood information. Furthermore, the comparison on SWRC to RESCAL and MW provides again insights into the improvements of ARE over these methods.

Please note that the construction of the oracle tensor M is described for each experiment in the respective subsections on p.7 and p.8.

Furthermore, the oracle tensor M is in all experiments identical for the MW model and ARE. The MW model can therefore be regarded as a baseline for how informative the employed heuristics are for a particular data set. We realized that this information is currently missing from the paper and we will add it to the final version. Thank you for pointing us to this.

Reviewer 3:
Thank you for your comments. We agree that a longer introduction to tensors would be preferable. As you kindly noted, due to space constraints we were only able to include those tensor concepts that are absolutely necessary for the paper. For this reason we also included a reference to the excellent review paper [Kolda, 2009], which provides a compact and accessible introduction to tensors.
Motivated by your comment, we will include further material on tensor unfolding and the tensor-times-matrix product in the supplementary material, such that these additional concepts are covered in the extended version of the paper.